# Mergeable nervous systems for robots

Nithin Mathews[1], Anders Lyhne Christensen[2], Rehan O'Grady[1], Francesco Mondada[3] & Marco Dorigo[1]

Robots have the potential to display a higher degree of lifetime morphological adaptation than natural organisms. By adopting a modular approach, robots with different capabilities, shapes, and sizes could, in theory, construct and reconfigure themselves as required. However, current modular robots have only been able to display a limited range of hardwired behaviors because they rely solely on distributed control. Here, we present robots whose bodies and control systems can merge to form entirely new robots that retain full sensorimotor control. Our control paradigm enables robots to exhibit properties that go beyond those of any existing machine or of any biological organism: the robots we present can merge to form larger bodies with a single centralized controller, split into separate bodies with independent controllers, and self-heal by removing or replacing malfunctioning body parts. This work takes us closer to robots that can autonomously change their size, form and function.

[1] IRIDIA, CoDE, Université Libre de Bruxelles, Brussels 1050, Belgium. [2] Instituto Universitário de Lisboa (ISCTE-IUL), Instituto de Telecomunicações, Lisbon 1649-026, Portugal. [3] Institut de Microinformatique (IMT), Faculté des Sciences et Technique de l'Ingénieur (STI), Ecole polytechnique fédérale de Lausanne, Lausanne 1015, Switzerland. Correspondence and requests for materials should be addressed to M.D. (email: mdorigo@ulb.ac.be)

The body shape and the structure of the control system of autonomous robots are typically specified at design time and remain constant throughout the robots' lifetime[1]. Most commonly, designers opt for a signaling and decision-making architecture in which sensors and actuators are connected to a central processing unit. We refer to this architecture as the robot nervous system, as it gives robots the ability to integrate sensory inputs and to coordinate actuators, analogous to nervous systems found in higher order animals. However, the nervous systems of existing robots are currently strictly mapped to their morphologies—even more strictly than biological nervous systems are mapped to body morphology in animals[2,3]. Contrary to animals, robots have the potential to display a high degree of morphological flexibility in which separate robotic units connect to one another to form new robots of different shapes and sizes[4–6]. However, the behavioral control paradigm adopted for current modular robots[4] resembles the biochemical signaling used by simple natural organisms that are able to change their body composition, such as the unicellular slime mold[7,8]. Much like their biological counterparts, current modular robots lack the essential ingredient that enables complex sensorimotor responses in higher order animals, namely a nervous system that spans the whole body and transforms a composite system into a single, holistic entity. Instead, the robotic units remain individually autonomous and rely on distributed approaches for coordination[9,10]. Sensorimotor coordination in current modular systems is thus limited or absent, which prevents them from solving tasks with the precision and reactivity provided by monolithic robots.

Here, we present mergeable nervous system (MNS) robots. An MNS robot is composed of one or more robotic units connected via the robot nervous system. We refer to the robotic unit responsible for the centralized decision making as the brain unit. Our MNS robots can adapt their bodies in two dimensions during task execution by splitting and merging to become new independent robotic entities of different shapes and sizes (Fig. 1). MNS robots split and merge their robot nervous system to retain sensorimotor coordination regardless of shape and size. MNS robots thus constitute a new class of robots with capabilities beyond those of any existing machine or biological organism: an MNS robot can split into separate autonomous robots each with an independent brain unit, absorb robotic units with different capabilities into its body, and self-heal by removing or replacing malfunctioning body parts—including a malfunctioning brain unit.

## Results

**Morphology-independent sensorimotor coordination**. For over 10 years[11,12], we have been developing the basic technologies that are a prerequisite for MNS robots. We have developed robotic units that can autonomously form physical connections with each other (Supplementary Fig. 1). Previously, however, each unit was always an independent robot[5]. While advances have been made in the development of algorithms and hardware that allow modular systems to form collective robot bodies of previously unattainable scales[13], the coordination and control paradigms for such robots are constrained to a predefined set of morphologies[14]. Our MNS robots are the first self-assembling multirobot system able to display sensorimotor coordination equivalent to that observed in monolithic robots. To demonstrate the capability of MNS robots to display morphology-independent sensorimotor coordination, we set-up an experiment in which we manually design the behavioral rules for ten robotic units so that they form a series of MNS robots of different shapes and sizes (Fig. 2 and Supplementary Movie 2). The different MNS robots all display the same coordinated sensorimotor reaction to a provided stimulus. This reaction involves pointing at the stimulus using light emitting diodes (LEDs), and retreating from the stimulus if it is sufficiently close. When a composite MNS robot points to the stimulus, only the LEDs closest to the stimulus illuminate, independently of the robotic unit to which those LEDs belong. When moving away from the stimulus, movements of all wheel actuators on all constituent robotic units are coordinated through the robot nervous system of the MNS robot, allowing smooth motion of the composite body (see Methods section).

**Body representation**. The physical connection topology of an MNS robot is a rooted tree. The logical topology of an MNS robot's nervous system follows the physical connection topology with each constituent robotic unit maintaining a recursive body representation of itself and all of its child robotic units. The representation includes the relative positions and hardware configurations of its child robotic units stored as a set of spatial

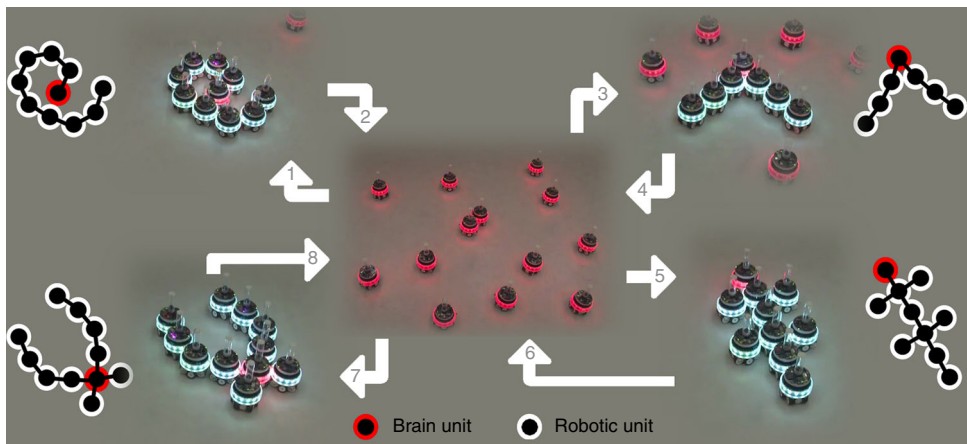

**Fig. 1** The mergeable nervous system concept. MNS robots are able to physically connect to one another and thereby merge into larger MNS robots of different shapes and sizes. In step 1, MNS robots consisting of a single robotic unit (*center*) self-assemble into a larger spiral-shaped MNS robot with a single brain unit (*upper left corner*). In step 2, the MNS robot splits and each of its robotic units becomes a one-unit MNS robot. The process is repeated three times (steps 3–8) during which the MNS robots merge into three larger MNS robots with different shapes (Supplementary Movie 1 for a video recording). The schematics show the brain unit (*in red*) and robotic units of each of the larger MNS robots and their merged robot nervous systems

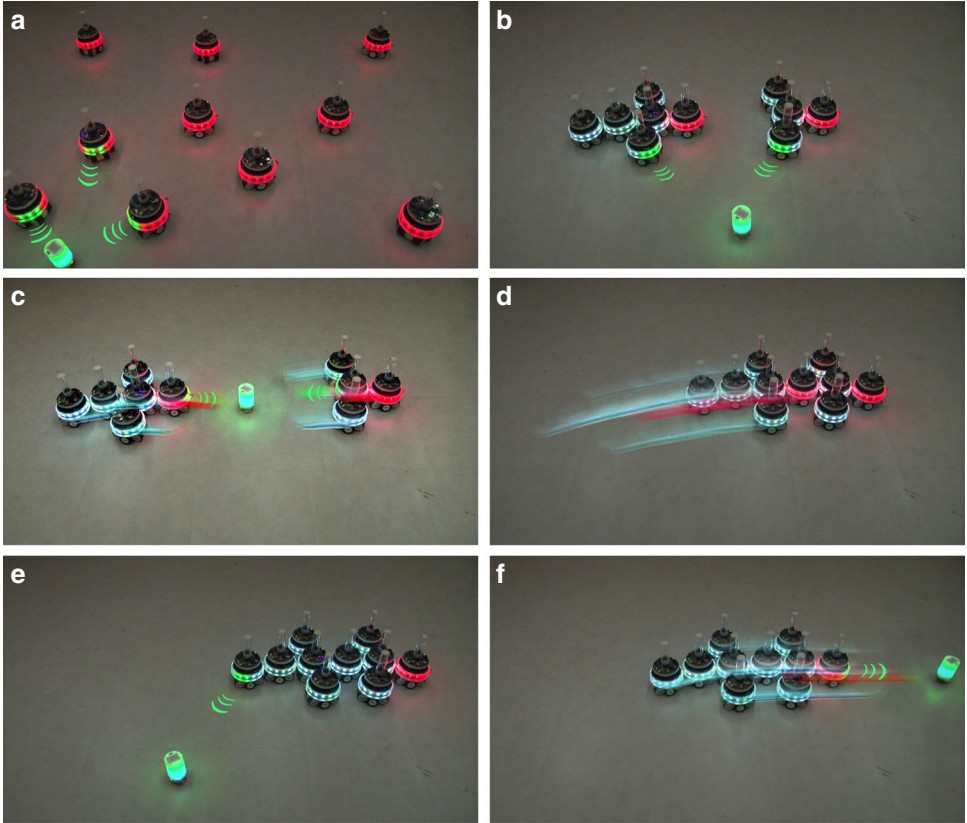

**Fig. 2** Morphology-independent sensorimotor coordination. Independently of shape and size, MNS robots display consistent sensorimotor reactions to a stimulus, while autonomously merging their bodies and robot nervous systems. Photos are snapshots from a single experiment (Supplementary Movie 2) in which ten MNS robots respond to a moving stimulus. We design the behavioral rules for the robotic units so that when the *green* stimulus enters a robot's sensor range, the robot 'points' at the stimulus by illuminating its three closest *green* LEDs (in a composite MNS robot, these are the closest LEDs on the closest constituent robotic unit). For clarity, we have added concentric *green lines* as overlays to highlight the pointing direction, and brain units have their LEDs illuminated in *red*. When the stimulus is 'too' close (i.e., proximity to any part of the MNS robot's body exceeds a threshold), the robot retreats from the stimulus. **a** Each robotic unit is an independent MNS robot in its own right. The stimulus provokes a reaction in three of the robots. **b** The ten MNS robots have merged to form two larger MNS robots. These newly formed robots are composite MNS robots each consisting of several robotic units. However, the two MNS robots are independent robots in their own right—they each have a single brain unit and a single robot nervous system. Both robots point at the stimulus. **c** The two MNS robots retreat from the stimulus. **d** The two MNS robots autonomously merge to form a larger MNS robot with a single brain unit. **e** The newly formed 10-unit MNS robot points at the stimulus. **f** The MNS robot retreats from the stimulus

relationships—the geometry of a robotic unit, and the physical arrangement of sensors and actuators (Supplementary Fig. 2). The recursive body representation allows MNS robots to react to radical morphological changes promptly. For instance, when an MNS robot splits into multiple robots with separate bodies, each root unit of the uncoupling body segments already has all the knowledge it needs to become the brain unit of the new independent robot. Given the tree structure of a mergeable nervous system, the root unit in a robot can always be identified unambiguously and serves as the brain unit. An important feature of morphology change in MNS robots is the speed at which the internal representation can be updated. When a merge between two MNS robots occurs, only a single message needs to be passed up the merged nervous system from the connecting MNS robot to the brain unit of the MNS robot to which it connects. The information contained in the message is incrementally updated by each intermediate unit with local topological information (Fig. 3), and the newly formed MNS robot incorporates all the sensing, actuation and computational capabilities of the units in the new body (Supplementary Movie 3). When an MNS robot changes shape or size, there is thus no need for time-consuming processes such as self-discovery[14], trial-and-error[15], or hormone-based messaging[16].

**Control logic**. Writing the control logic for a composite MNS robot is an entirely new challenge. It is impractical for such logic to take into account every possible morphology, i.e., relative placement of sensors, actuators, and body parts. Our solution is to divorce the control logic from the morphology and from individual sensors and actuators. The control of an MNS robot is expressed in high-level logic that is independent of the size and shape of the robot (Supplementary Table 1 for a list of the commands available to MNS robots). The brain unit issues high-level commands that are propagated through the robot nervous system. If a high-level command applies to a robotic unit, the robot nervous system locally translates the command into instructions for the unit's actuators. Figure 4 shows how responsibility is delegated as part of the information flow in the robot nervous system of a merged robot.

Spatial coordination is achieved by enforcing coordinate translation every time a sensor or actuator message is passed from robotic unit to robotic unit. As each robotic unit knows the relative location of its parent robotic unit and child robotic units, spatial references are translated into the frame of reference of the receiving unit before the messages are transmitted. In this way, units always receive messages that are meaningful within their own frame of reference. The coordination of the actuators on

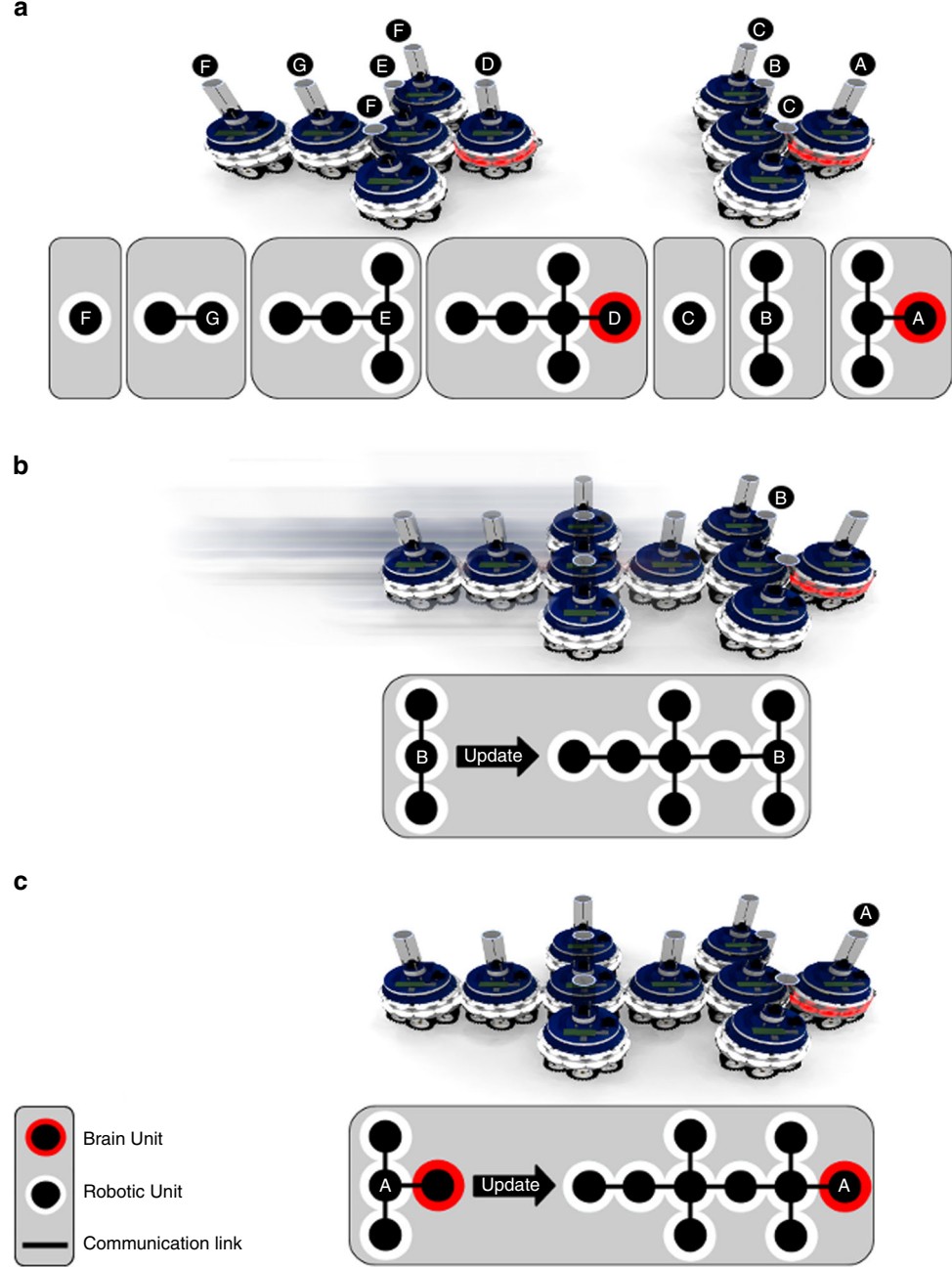

**Fig. 3** Propagation of internal representation during the merging of two MNS robots. **a** The internal representations of robotic units are shown in the insets. In each of the insets, the robot whose internal representation is illustrated is indicated by the corresponding letter. Note that units that are not the brain unit only have awareness of their descendant units. **b** The robot on the *left* has assembled to the robot on the *right*. The brain unit of the robot that is attaching (*left* robot) cedes authority to the brain unit of the robot to which it is attaching (*right* robot). The brain unit of the new merged robot still does not have an accurate internal representation of its new morphology. However, its child robotic unit has already updated its internal representation. **c** Information about the morphology has propagated to the brain

different robotic units takes any propagation delay associated with the robot nervous system into account. The length of the delay is a function of the path from the brain unit to the most distant leaf unit in a morphology, and the communication technology on which the MNS is based, Wi-Fi in our case. If the propagation delay is significant, actuator instructions are not executed by a robotic unit immediately upon reception, but instead postponed until the instructions have had time to propagate to all units in the body. In Supplementary Fig. 3, we provide a detailed example of how spatial and temporal actuator coordination happen in an MNS robot.

**Scalability**. We consider scalability with respect to the number of constituent robotic units in an MNS robot, first in terms of computational resources required to control the robot and then in terms of reaction time, that is, the time required for a robot to react to a new stimulus.

In an MNS robot, sensory data from child robotic units are fused by their parent robotic unit before being passed up the robot nervous system towards the brain unit. The computational cost of sensor data extraction and processing for any robotic unit, including the brain unit, is thus proportional to the number of immediate child robotic units, rather than a function of the total

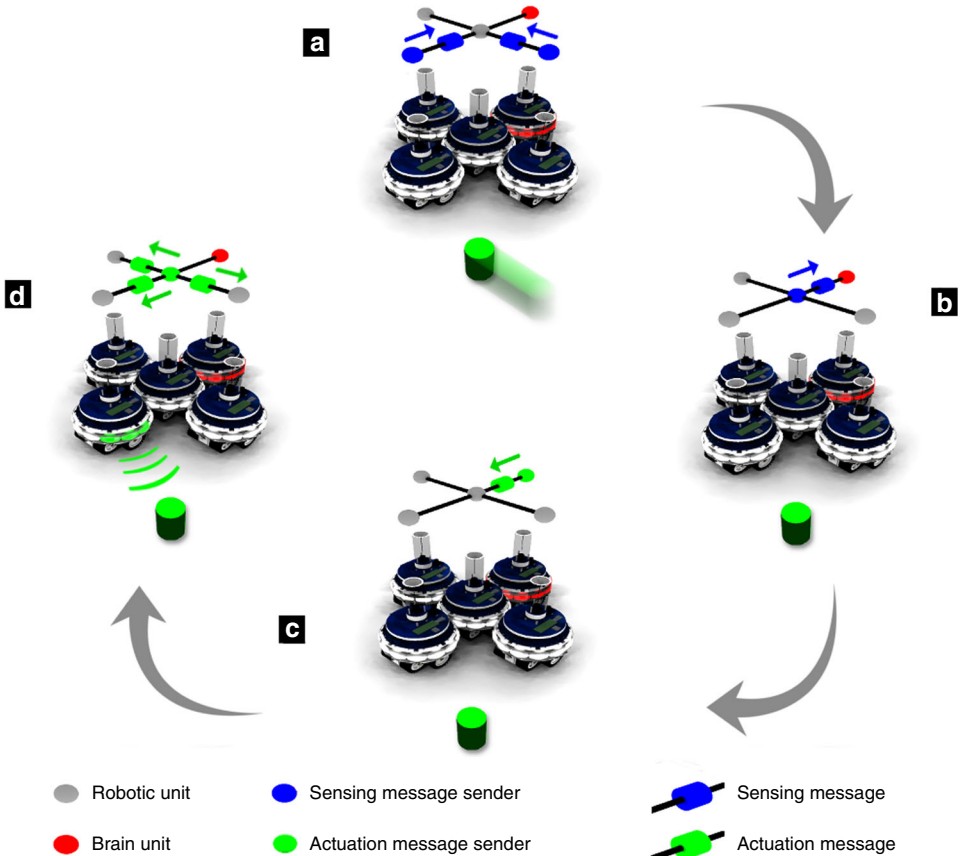

**Fig. 4** Sensor and actuator information flow in an MNS robot. A 5-unit MNS robot detects and then responds to a stimulus. a: The stimulus moves within sensor range of two robotic units in the MNS robot. Both robotic units perform the computationally intensive visual image processing required to analyze their camera feeds. They pass an abstraction of this information (e.g., existence and coordinates of the stimulus) up to their parent robotic unit (i.e., the unit in the center of the robot) using a Wi-Fi connection. b: The parent robotic unit fuses the information coming from two of its child robotic units, to form a more accurate estimate of the stimulus' coordinates. The parent robotic unit then passes this single item of information up to its own parent robotic unit, which in this case is the brain unit. The brain unit decides what action to take, based on the data it has received—in this case, the decision is to point at and retreat from the stimulus. c: The brain unit issues high-level actuator commands. d: The high-level commands are translated into actuator instructions individually by each robotic unit

number of robotic units in an MNS robot's body. Scalability issues would quickly arise if all sensory data (such as camera feeds) were collected and processed in the brain unit. Instead, the use of local sensor fusion allows the MNS method to scale gracefully in terms of computation.

For an MNS robot to react to a new stimulus, a message must propagate from the constituent robotic unit sensing the stimulus to the brain unit, which in turn must propagate a message detailing the response back through the body. The reaction time for an MNS robot thus depends not only on the number $k$ of constituent robotic units, but also on the MNS robot's shape (the connection topology of the constituent robotic units). The worst case reaction time is given by $2lp \times \tau$, where $lp$ is the length (in robotic units) of the path from the brain unit to the most distant robotic unit in the body, and $\tau$ is the communication delay between two adjacent robotic units. In our current implementation, adjacent robotic units can exchange messages every $\tau = 100$ ms. The 4-unit MNS robot shown in Fig. 2b, c has a longest path of $lp = 2$ robotic units and its reaction time is thus $2 \times 2$ unit $\times$ 100 ms unit$^{-1}$ = 400 ms. Similarly, the 6-unit MNS robot in Fig. 2b, c has $lp = 3$ units and therefore a worst case reaction time of 600 ms, while the 10-unit MNS robot in Fig. 2d–f has $lp = 5$ units and thus a worst case reaction time of one second. In general, the reaction time of an MNS robot of size $k$ falls between two extremes depending on body shape: the upper-bound

corresponds to a robot with a linear shape for which the reaction time is $2k \times \tau$, while the lower-bound corresponds to a shape that is as compact as possible. Given the circular shape of our robotic units, the most compact shape possible is one in which constituent robotic units are arranged in a hexagonal lattice pattern around the brain unit, which yields a longest path of approximately $lp = \log_2 k$ units and therefore a reaction time of 2 $\log_2 k \times \tau$.

**Self-healing**. MNS robots can, in principle, self-heal by combining their splitting and merging capabilities to substitute faulty components, including their brain unit. After a fault has been detected (see Methods section), MNS robots can reconfigure their bodies to excise the faulty robotic units and possibly substitute them with new, spare units. To demonstrate the self-healing capability, we design behavioral rules for eight robotic units so that they self-assemble into an MNS robot with an Y-shape. We then inject a fault, first in the brain unit of the MNS robot and then in a robotic unit of the MNS robot body. In the first part of the experiment, in which the brain unit fails, the three child robotic units detect the fault and respond by detaching, thereby creating three new, independent MNS robots each with a brain unit of its own. The three robots then merge with one another to form a new, larger robot with a morphology as close as possible to the original robot (Fig. 5 and first segment of

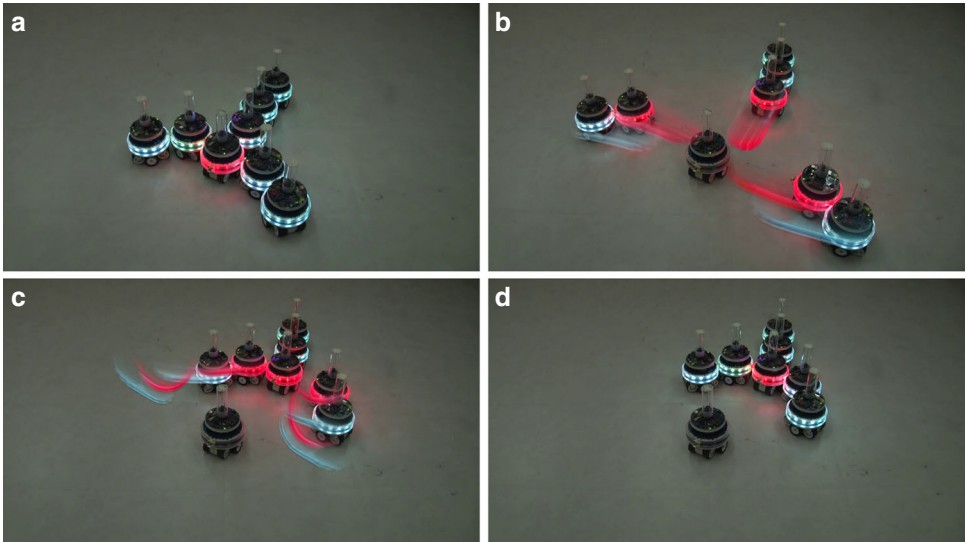

**Fig. 5** An MNS robot self-heals after its brain unit develops a fault. Photos are snapshots from an experiment (Supplementary Movie 4). **a** A fault is injected in the brain unit. Using a heartbeat protocol (see Methods section), the three robotic units attached to the brain unit detect the fault. **b** All robotic units attached to the faulty brain unit detach. Each of these units now becomes the brain unit of a new MNS robot. **c, d** The three new MNS robots merge to recreate a single composite MNS robot with a single brain unit

Supplementary Movie 4). In the second part of the experiment, a robotic unit in the body of the MNS robot fails. In this case, the body part containing the failed unit is detached from the MNS robot and two new robotic units are recruited to recreate the original robot (Supplementary Fig. 4 and second segment of Supplementary Movie 4). In general, reconfiguration sequences in response to faults are task and morphology specific. The simplest fallback case is for all of the child robotic units to completely disassemble, and to reform another morphology from scratch, but partial disassembly may be sufficient in many cases.

## Discussion

In this paper, we have demonstrated MNS robots able to form bodies of different shapes and sizes while retaining fine sensorimotor coordination. Our robots currently operate in two dimensions and are limited to rigid connections between the constituent robotic units. In future work, we intend to extend the MNS concept to self-reconfigurable modular robots that operate in three dimensions and with flexible joints. Building on the MNS method, robots of the future will display a new type of adaptivity by autonomously choosing appropriate morphologies for the tasks and environments they encounter. Understanding which morphology is appropriate to which task and environment is a problem nature solves over millions of years using evolution. To solve the same problem on the fly, we might be able to rely on ever increasing computing power and advances in evolutionary computation techniques[17, 18]. Our vision is that, in the future, robots will no longer be designed and built for a particular task. Instead, we will design composable robotic units that give robots the flexibility to autonomously adapt their capabilities, shape and size to changing task requirements.

## Methods

**Internal body representation**. In an MNS robot, each constituent robotic unit maintains an internal representation of itself and all of its child robotic units, including the placement of individual sensors and actuators. We use a predefined set of templates for robotic units, where each template corresponds to the hardware configuration of a specific type of robotic unit (in our case the different modular arrangements of the marXbot platform, see subsection Robot hardware and self-assembly below). In Supplementary Fig. 2, two MNS robots formed with robotic units of two different hardware configurations are shown. This ability to include information about malfunctioning sensors and actuators is important to allow a

composite MNS robot to be fault tolerant—the larger the MNS robot, the higher the likelihood of partial failures, and the more important it is that the robot is aware of, and can compensate for such failures. The internal body representation thus includes information about the hardware configuration of each robotic unit, how robotic units are connected to one another, and information about any malfunctioning sensors and actuators. The representation is recursive and starts with information about the brain unit's hardware configuration, any malfunctions, and the number of child robotic units directly connected to it. The representation then contains information about each child robotic unit's hardware configuration, at what angle it is connected to its parent robotic unit, any malfunctions, and information about each of its child robotic units, and so on. When robotic units communicate body representations, for instance during a merge between two MNS robots, we serialize the body representation using the following format (in Backus-Naur form):

```
<MNS>::=<hardware-configuration>,<connection-angle>,
<malfunctioning-sensors>*,  <malfunctioning-actuators>*,
<num-children>,(<MNS>)*
```

In Supplementary Fig. 2, the serialized versions of the internal body representations for the two MNS robots in the figure are shown at the bottom of each pane.

**Robot hardware and self-assembly**. To enable self-assembly and to study the MNS approach, we developed the marXbot robotic platform[11]. Each unit has a circular chassis with a diameter of 17 cm (Supplementary Fig. 1). A combination of tracks and wheels provides the marXbots with differential drive motion capabilities. The marXbot is fully autonomous and is equipped with an ARM 11 processor (i.MX31 clocked at 533 MHz and with 128 MB RAM) running a Linux-based operating system. Onboard sensors and actuators include 12 RGB-colored LEDs distributed around its chassis, and an omni-directional camera mounted in a perspex tube on the robot's turret. As shown in Supplementary Fig. 1, the marXbot can be configured with different sensors and actuators.

An inter-robot connection module enables one marXbot to form a physical connection with another marXbot. This module is composed of an active gripping device with three fingers together with a passive docking ring (Supplementary Fig. 5). A marXbot forms a physical connection with another marXbot by inserting its three fingers into the target marXbot's docking ring and then opening its fingers. One marXbot can connect to another marXbot anywhere except at its gripper (gripper–gripper connections are impossible), resulting in a gripping target perimeter of 320° around the marXbot's body.

To enable self-assembly, marXbots must be able to locate each other, and then the attaching robot must be guided to connect at an appropriate location on the robot receiving the connection[19]. The communication used by the marXbots must thus be situated[20]. In situated communication, the relative location of the sender can be estimated by the receiver. We recently developed the range-and-bearing communication module (mxRAB) that enables situated communication on the marXbot[21]. The mxRAB device provides situated communication at 10 Hz and up

to 5 m with a combination of infrared and radio technologies. Note that the mxRAB device is only used during the self-assembly process. Communication between physically connected units in an MNS robot happens through Wi-Fi.

**Fault detection**. The detection of total failure of robotic units is accomplished through a heartbeat protocol[22]. Heartbeats, that is, periodic signals sent to indicate normal operation, are generated by the brain unit and sent through the robot nervous system at a fixed frequency. The absence of a heartbeat from a parent robotic unit tells a child robotic unit that its parent unit is faulty, while the absence of an acknowledgment from a child robotic unit tells the parent unit that its child unit is faulty. Timing offsets of the heartbeat-checking window ensure that multiple descendant robotic units do not react to the failure of one ancestor robotic unit (the offset gives the child robotic unit of a faulty robot time to become the brain unit of the sub-morphology and start generating heartbeats). The anatomy of a heartbeat message is shown in Supplementary Table 1. When the brain unit sends a heartbeat message, it includes information about the expected frequency of heartbeats, and what actions body parts should take if a fault is detected. Each robotic unit in the MNS robot acknowledges each heartbeat received from its parent robotic unit, and then propagates the heartbeat to its own child robotic units.

In MNS robots, it is the responsibility of each robotic unit to detect partial failures in itself. The MNS method is agnostic as to what method a robotic unit uses to detect its own partial failures[23]. The detection of a partial failure triggers a series of internal representation update messages (exactly the same type of message that is used to update the internal representation after a merge or a split, Fig. 3). Through these messages, ancestor robotic units can update their internal representations with the knowledge of the partial failure.

**Data availability**. All data generated or analyzed during this study are included in this published article (and its Supplementary Information files). The computer code to control the robots is available from the authors upon request.

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

## Acknowledgements

This work was supported by the European Research Council through the ERC Advanced Grant "E-SWARM: Engineering Swarm Intelligence Systems" (contract 246939) to Marco Dorigo. Nithin Mathews acknowledges support from Wallonia-Brussels-International (WBI) through a Scholarship for Excellence grant. Rehan O'Grady and Marco Dorigo acknowledge support from the Funds for Scientific Research F.R.S.-FNRS of Belgium's French Community, of which they are a scientific collaborator and a research director, respectively. Anders Lyhne Christensen acknowledges support from Fundação para a Ciência e a Tecnologia (FCT) through grant UID/EEA/50008/2013.

## Author contributions

A.L.C. conceived and formulated the initial MNS idea; N.M. implemented the robot controllers and carried out initial evaluations. F.M. and M.D. coordinated the design of the hardware. N.M., R.O. and M.D. designed the experiments which were then performed by N.M. All authors discussed the results and wrote the manuscript.

## Additional information

**Competing interests:** The authors declare no competing financial interests.

**Change history:** A correction to this article has been published and is linked from the HTML version of this article.

