## [Peer Review File · Nature Communications]

Reviewers' comments:

Reviewer #1 (Remarks to the Author):

The core contribution of this paper is how to coordinate a group of mobile robots to performed coherently as a single entity with no dependency on the spatial organization of the constituent robots. I think this specific contribution is thought provoking and useful for the community. However, I think the paper needs to be revised based on the comments below.

1) Scalability. Clearly the authors will agree that scalability of the proposed algorithm is important. The reaction time of a robot entity is proportional to the longest path in the robot times the robot-to-robot communication delay. Hence, as the robot entity grows larger its reaction time goes down. I think this is an important down-side to mention and discuss. I actually pointed this out here and never made a come-back:

Støy, K.;Shen, W.-M.;Will, P., On the Use of Sensors in Self-Reconfigurable Robots, Proc., 7th Int. Conf. on The Simulation of Adaptive behavior, pp. 48-57, Edinburgh, UK, 2002.

2) The use of the term modular robots. I can see why you view your system as a modular robot. Clearly, it consists of independent modules which connect physically. However, I feel it lacks the potential to move into the third dimension. I think this is important because the complexity of the coordination task is simpler in a 2d system than in a 3d system. E.g. in a 3d system you might be doing legged locomotion or manipulation. If you do these tasks it is important that the whole system stays coordinated at all times. Otherwise, the robot may endanger itself or its surroundings. E.g. you cannot stop a leg in mid-air and wait for the system to coordinate to proceed into a jump. I would be more comfortable if you call the system a swarm robot system or multi-robot system because you have the huge advantage that all modules can just stop without risking any damage. There is also the pesky question of 3d self-reconfiguration which is not the same as in 2d..

3) Line 1 gave the paper a bad start. "Robots have the potential to be much more adaptable than natural organisms.". While the word "potential" means this could happen far in the future I think we have to look at what is within reach today. E.g. Nature still has a few things up its sleeve that is hard for us to do e.g. self-replication which clearly is a useful factor for the adaptability of bacteria colonies. I suggest to tone it down a bit and say something about "life-time morphological adaptation" although slime-molds of course have that nailed already.

4) The experimental support could be better. The paper only include proof-of-concept experiments. In particular it would be nice with data on how the system scales with the size of the robot entity.

5) Nature Comms asks me to comment on the possibility to reproduce your work. I think the high-level algorithm which is the important bit is. However, of course the algorithm is highly tailors to your specific robot limiting its general usability, but I think that is how it has to be in robotics.

Future work:

1) It would seem that if you know what task you do you can optimize communication for this. E.g. only passing sensor information from the front to the back of robot. If you know spatial organization even more.. Maybe something for future work. Maybe something that can be learned.

2) I think for modular robots you would need a two-tier approach. An underlying system that maintains coordination and a task switching mechanism that your current system could be an answer

to.

Overall, I think the work is very good and if you address the relative minor comments and make some more experimental support this will be an important paper moving forward.

Kasper Stoy, Ph.D.
Professor
IT University of Copenhagen

Reviewer #2 (Remarks to the Author):

. This is an innovative manuscript that addresses the subject of reconfiguration in robots. It unfortunately conflates neuronal networks and stigmergy although it never mentions the concept of stigmergy, that is a key organizational concept of social insect societies.

. In the nervous system, reconfiguration is feasible, but it uses an entirely different mechanism than that proposed here. In the CNS, the topology of synaptic networks is largely invariant. Circulating neuromodulators alter instead the properties of individual neurons and the synaptic strengths between connected elements to reconfigure the CNS networks for different functions. Thus concept of minibrains connecting and disconnecting proposed for reconfiguration is without basis in nature.

. The phenomena described here is more similar to stigmergy and the word stigmergy does not appear in the manuscript. Moreover stigmergy is based largely on indirect cues, predominately olfactory. The fundamental neuronal architecture of stigmergy and indeed all innate behavior in all organisms is based on command neurons, coordinating neurons and central pattern generators adapted by phase and amplitude modulating reflexes. These innate networks are subject to a broad variety of neuromodulators to produce a multiplicity of reconfigured behavior using the same network topologies.

. I found much of the behavior of these systems to be inadequately explained. Vide. "When moving away from the stimulus, movements of all wheel actuators on all constituent units are coordinated through the nervous system of the MNS robot, allowing smooth motion of the composite body (see Methods)." My search of the methods yielded no such coordinating mechanism. Moreover the source of spontaneous directed behavior by individual robots appears unexplained. The latching mechanism requires a planar substrate and any obstructions will block the light based interactions. Thus the interactions will only occur in a specific environment. Moreover, I found no clear explanation of how the final configuration of a robot is specified and how this relates to any specific goal.

REVIEWERS' COMMENTS:

Reviewer #1 (Remarks to the Author):

This is my second review of the paper. I feel that the critique I had to the first version has been appropriately handled by the authors and I suggest the paper is accepted.

Reviewer #2 (Remarks to the Author):

I found the authors to be unresponsive to my criticisms.

First: Their statement "we are not trying to imitate nature" contradicts their title "Mergable Nervous Systems.....". As i indicated. Their proposed architecture has nothing to do with the nervous system, yet they persist to use nervous system analogies throughout the manuscript. Try performing a search for the word brain and you will get the picture. My favorite is on line 208-210 with the statement "Heartbeat pulses are generated by the brain unit and sent through the robot nervous system at a fixed frequency."

Consider my comment: Moreover the source of spontaneous directed behavior by individual robots appears unexplained.

and their explanation:

- We have now clarified that robots are manually programmed: we have added the sentence "we programmed ten robotic units to form a series ..." in the main paper (line 50) and the sentence "We programmed the robotic units so that ..." in the caption of Figure 2.

It is not valid to consider manual programming as a mechanism for "self-assembling multi robots" (Lines 47-51)

Moreover, I agree with the statement (Line 80-82) "Writing the control logic for a composite MNS robot is an entirely new challenge. It is impractical for such logic to take into account every possible morphology, i.e., relative placement of sensors, actuators and body parts." However, In Supplementary Fig. 3, we provide a detailed example of how spatial and temporal actuator coordination happen in an MNS robot. (Line 100-101) does not include such an explanation.

Answers to referees' comments for paper titled "Mergeable Nervous Systems for Robots"

by N. Mathews, A. L. Christensen, R. O'Grady, F. Mondada, and M. Dorigo

Please note that the revised version of our paper contains two types of changes. First, there are changes implemented as a result of the reviewers comment. These changes are highlighted using a red font in the paper. Second, the text has been modified and rearranged to conform the paper to the Nature Communications formatting guidelines (the first submission of the paper was to Nature which has different format requirements; Nature editors suggested transfer to Nature Communications and the transfer was done without any reformatting).

Reviewer #1

The core contribution of this paper is how to coordinate a group of mobile robots to performed coherently as a single entity with no dependency on the spatial organization of the constituent robots. I think this specific contribution is thought provoking and useful for the community.

- We thank the reviewer for the positive feedback.

1) Scalability. Clearly the authors will agree that scalability of the proposed algorithm is important. The reaction time of a robot entity is proportional to the longest path in the robot times the robot-to-robot communication delay. Hence, as the robot entity grows larger its reaction time goes down. I think this is an important down-side to mention and discuss.

- In the Results section in the main paper, we have introduced a new subsection on scalability (starting on line 102), in which we discuss the relation between MNS robot size, shape and reaction time (lines 113-130).

2) The use of the term modular robots. I can see why you view your system as a modular robot. Clearly, it consists of independent modules which connect physically. However, I feel it lacks the potential to move into the third dimension. I think this is important because the complexity of the coordination task is simpler in a 2d system than in a 3d system. E.g. in a 3d system you might be doing legged locomotion or manipulation. If you do these tasks it is important that the whole system stays coordinated at all times. Otherwise, the robot may endanger itself or its surroundings. E.g. you cannot stop a leg in mid-air and wait for the system to coordinate to proceed into a jump. I would be more comfortable if you call the system a swarm robot system or multi-robot system because you have the huge advantage that all modules can just stop without risking any damage. There is also the pesky question of 3d self-reconfiguration which is not the same as in 2d.

- We are now explicit about the fact that our system operates in two dimensions (lines 31-32 and 151-152), and when we refer to our system, we call it a "self-assembling multirobot system" (line 47). We also mention the extension of the MNS method to robots operating in three dimensions as future work.

For the sake of discussion, we believe that our MNS method is extendable to robots operating in three dimensions. For instance, there would be no need for a robot that has to jump to stop a leg in mid-air and wait for the system to coordinate before a subsequent action is taken: when a sequence of actions is known a priori, it can be queued up in the MNS to avoid any delay between actions. In case the robot decides to jump while performing another action, the high-level command that triggers the jump could be propagated while the robot continues to perform its current action, and it would thus not have to stop with a leg in mid-air. Furthermore, an

equivalent of reflexes common in biological nervous systems could be implemented in the MNS method by giving individual body parts the responsibility to, for instance, ensure stability. In the current MNS design, robotic units are already responsible for some local, low-level control: when a body grows, it is the responsibility of the robotic unit to which a new body part connects to coordinate the physical attachment without the brain unit needing to intervene. Because we do not have any results to support our reasoning above, we prefer not to discuss any of these points in the paper, but limit ourselves to say that we intend to extend MNS to robots operating in three dimensions in future work (lines 152-154).

3) Line 1 gave the paper a bad start. "Robots have the potential to be much more adaptable than natural organisms.". While the word "potential" means this could happen far in the future I think we have to look at what is within reach today. E.g. Nature still has a few things up its sleeve that is hard for us to do e.g. self-replication which clearly is a useful factor for the adaptability of bacteria colonies. I suggest to tone it down a bit and say something about "life-time morphological adaptation" although slime-molds of course have that nailed already.

- The first sentence of the abstract (lines 1-2) has now been rewritten:

"Robots have the potential to display a higher degree of lifetime morphological adaptation than natural organisms."

4) The experimental support could be better. The paper only include proof-of-concept experiments. In particular it would be nice with data on how the system scales with the size of the robot entity.

- As stated in our response to point 1, we have added an entirely new subsection in Results titled "Scalability" (lines 102-130). In the new subsection, we provide additional data about the robot-to-robot communication delay in our experiments and the reaction times for different sizes of the robot entity, corresponding to the MNS robots shown in Figure 2.

5) Nature Comms asks me to comment on the possibility to reproduce your work. I think the high-level algorithm which is the important bit is. However, of course the algorithm is highly tailors to your specific robot limiting its general usability, but I think that is how it has to be in robotics.

- To simplify the work of anyone interested in reproducing our work, we have made our computer code available upon request. We have added a statement at the end of the Methods section (line 230).

Future work:

6) It would seem that if you know what task you do you can optimize communication for this. E.g. only passing sensor information from the front to the back of robot. If you know spatial organization even more.. Maybe something for future work. Maybe something that can be learned.

- We agree that there are several interesting opportunities for task-, situation-, and shape-dependent communication optimization in MNS robots. However, since we have not implemented and experimented with any such optimizations, we prefer not to discuss them in the paper.

7) I think for modular robots you would need a two-tier approach. An underlying system that maintains coordination and a task switching mechanism that your current system could be an answer to.

- As we discussed in our answer to point 2 above, we believe that our MNS method can be used

not only as a task switching mechanism, but also as a coordination mechanism.

Overall, I think the work is very good and if you address the relative minor comments and make some more experimental support this will be an important paper moving forward.

- We have now addressed the comments made by the reviewer and provided additional data on the reaction time for the different MNS robots in the experiment shown in Figure 2 with up to 10 robotic units (see Results lines 113-130).

Reviewer #2

This is an innovative manuscript that addresses the subject of reconfiguration in robots.

- We thank the reviewer for the positive feedback.

It unfortunately conflates neuronal networks and stigmergy although it never mentions the concept of stigmergy, that is a key organizational concept of social insect societies. In the nervous system, reconfiguration is feasible, but it uses an entirely different mechanism than that proposed here. In the CNS, the topology of synaptic networks is largely invariant. Circulating neuromodulators alter instead the properties of individual neurons and the synaptic strengths between connected elements to reconfigure the CNS networks for different functions. Thus concept of minibrains connecting and disconnecting proposed for reconfiguration is without basis in nature. The phenomena described here is more similar to stigmergy and the word stigmergy does not appear in the manuscript. Moreover stigmergy is based largely on indirect cues, predominately olfactory. The fundamental neuronal architecture of stigmergy and indeed all innate behavior in all organisms is based on command neurons, coordinating neurons and central pattern generators adapted by phase and amplitude modulating reflexes. These innate networks are subject to a broad variety of neuromodulators to produce a multiplicity of reconfigured behavior using the same network topologies.

- In this paper, we are not trying to imitate nature. We are proposing a robot control architecture that allows self-assembling multirobot systems to achieve a degree of control on their body shape and behavior that was not possible before. Therefore, even though we believe the referee's comments are in general correct, they do not apply to our work, as we do not make any claims that the low-level functioning of our robotic nervous system bares any resemblance to that of nervous systems in animals.

Concerning the comment about stigmergy, it should be noted that the robotic units comprising an MNS robot communicate explicitly through Wi-Fi, and therefore our MNS robots do not rely on stigmergy. This is explained in the caption of Figure 4, where we now explicitly state that sensory information is transmitted from unit to unit "...using a Wi-Fi connection", in the Results subsection "Control logic" (lines 97-98), and in Methods (lines 204-206).

I found much of the behavior of these systems to be inadequately explained. Vide. "When moving away from the stimulus, movements of all wheel actuators on all constituent units are coordinated through the nervous system of the MNS robot, allowing smooth motion of the composite body (see Methods)." My search of the methods yielded no such coordinating mechanism.

- The coordination mechanism was explained in Methods, but has now been moved to the Results subsection titled "Control logic". The relevant part is the one starting with "Spatial coordination is achieved by ..." (lines 90-101). Also, please note that a detailed example of actuator coordination in an MNS robot is provided in Supplementary Fig. 3.

Moreover the source of spontaneous directed behavior by individual robots appears unex-

plained.

- We have now clarified that robots are manually programmed: we have added the sentence “we programmed ten robotic units to form a series ...” in the main paper (line 50) and the sentence “We programmed the robotic units so that ...” in the caption of Figure 2.

The latching mechanism requires a planar substrate and any obstructions will block the light based interactions. Thus the interactions will only occur in a specific environment.

- The referee is right to say that the latching mechanism requires a (semi-)planar substrate. However, this is not a major limitation of the approach we propose in this paper. Rather, it is due to the nature of the specific hardware that we have chosen to use. Future implementations of the ideas proposed in this article are not constrained by the (semi-)planar latching mechanism. In the last paragraph of the main paper, we now explicitly state that we intend to extend MNS to robots operating in three dimensions.

The referee’s sentence “any obstructions will block the light based interactions” makes us wonder if there wasn’t a misunderstanding which might explain the referee’s comment concerning stigmergy above. Interactions between units in an MNS robot happen via Wi-Fi (i.e., the messages sent around the robot nervous system use Wi-Fi connections between physically connected robots). This has now been clarified in the paper (see caption of Fig. 4, lines 97-98, and lines 204-206; see also answers to previous comments). The robots’ colored LEDs do not play any role in the MNS system described in this paper—the LEDs are used purely as a display mechanism to allow a human observer to see what is going on: green indicate which robotic unit in an MNS robot is the closest one to the external stimulus; red identify the brain of an MNS robot; and white identify the non-brain robotic units of an MNS robot.

Moreover, I found no clear explanation of how the final configuration of a robot is specified and how this relates to any specific goal.

- We specify the final configuration a priori (we already partially answered this point above, where we say that robots were programmed by us). We also added the sentence “To demonstrate this capability, we programmed eight robotic units so that they self-assemble into an MNS robot with an Y-shape” (lines 135-136) to clarify this point for the experiment shown in Figure 5.

Answers to comments for paper titled “Mergeable Nervous Systems for Robots”

by N. Mathews, A. L. Christensen, R. O’Grady, F. Mondada, and M. Dorigo

Editor

While reviewer #1 has been generally in favor of publication, reviewer #2 continuously raised a few questions that need to be addressed carefully in any revision.

- In the rest of this document, we provide detailed explanations about how we addressed your and reviewer #2’s comments.

In particular, you can see that this reviewer is skeptical about the use of biological terminology (brain, heartbeat, children etc) throughout the paper. Although we appreciate a tradition in the field of robotics to make such analogs only for the sake of discussion, we are inclined to agree with this reviewer that the similarities between robotics and real nervous systems are rather limited at best. Neither do their appearances in the discussion help understand the work - you need to elaborate on the experimental detail at many places. Having said that, we request to remove all the unjustified terminologies from the revision, including brain, heartbeat, children etc. You could continuously choose to name the system as mergeable nervous system if you insist. To this end, please emphasize in the main paper that it is used as metaphor other than having a scientific meaning.

- In order to emphasize that we use the terms nervous system and brain in a metaphorical way, we have added the following sentence at the beginning of the paper: “*Most commonly, designers opt for a signaling and decision-making architecture in which sensors and actuators are connected to a central processing unit. We refer to this architecture as the robot nervous system, as it gives robots the ability to integrate sensory inputs and to coordinate actuators, analogous to nervous systems found in higher order animals.*” (lines 15–19). Concerning terminologies:
 - **brain**: we have removed the term brain from the abstract to avoid confusion; in the main paper, we no longer use the term “brain” alone. Rather, we clearly define the term “brain unit” (lines 33–35) and we use it in the rest of the paper: “*An MNS robot is composed of one or more robotic units connected via the robot nervous system. We refer to the robotic unit responsible for the centralized decision making as the brain unit*” (lines 33–35).
 - **heartbeat**: we have explained what a heartbeat is: “*The detection of total failure of robotic units is accomplished through a heartbeat protocol.²² Heartbeats, that is, periodic signals sent to indicate normal operation, are generated by the brain unit and sent through the robot nervous system at a fixed frequency*” (lines 223–225). Please note that this is not our terminology: the heartbeat protocol is a well-known mechanism used to check for normal behavior in computer systems. We now provide a reference in the paper (reference 22).
 - **child, parent, descendant, ancestor, leaf**: We have added the sentence “*The physical connection topology of an MNS robot is a rooted tree*” (line 66), so that it is now clear that “child”, “parent”, “descendant”, “ancestor”, and “leaf” are used with their standard meanings in the context of graph theory. Also, the first time that any of these words occur in a sentence, we include “robotic unit” for clarity, so that “child” is now “child robotic unit” and so on.

Comments in the PDF files

- Below is a list of changes implemented in response to the comments in the PDF files not addressed elsewhere in this document or in the manuscript checklist:
 - Page 3: all our figures have a title as required.
 - Page 3: we have checked usage of present tense.
 - Page 11: we have added an explanation in human language of the internal body representation (lines 183–190). However, we prefer to also include the formal representation in machine script for precision.
 - Page 13: we have checked the references and used abbreviated journal titles.
 - We have implemented all the other low-level comments/corrections (including those concerning the supplementary material).

Reviewer #2

I found the authors to be unresponsive to my criticisms. First: Their statement "we are not trying to imitate nature" contradicts their title "Mergable Nervous Systems.....". As i indicated. Their proposed architecture has nothing to do with the nervous system, yet they persist to use nervous system analogies throughout the manuscript. Try performing a search for the word brain and you will get the picture.

- As explained above in our answer to the editor's comments, we have modified our paper to make it clear that we use the terms in a metaphorical way: we have added the sentence at lines 33–35, removed occurrences of the term "brain" in the abstract, and defined what we mean by "brain unit" in the beginning of the paper.

My favorite is on line 208-210 with the statement "Heartbeat pulses are generated by the brain unit and sent through the robot nervous system at a fixed frequency."

- As explained above, the heartbeat protocol is a well-known mechanism used for fault detection in distributed computer systems. We now provide a short explanation of the protocol (lines 223–225) and a reference (reference 22).

Consider my comment: Moreover the source of spontaneous directed behavior by individual robots appears unexplained. and their explanation: We have now clarified that robots are manually programmed: we have added the sentence we programmed ten robotic units to form a series ... in the main paper (line 50) and the sentence We programmed the robotic units so that ... in the caption of Figure 2.

It is not valid to consider manual programming as a mechanism for "self-assembling multi robots" (Lines 47-51)

- The spontaneous directed behavior of our self-assembling robots is the result of behavioral rules designed by us. This has now been further clarified, see for instance lines 53–57 in the main paper. It should be noted that, to date, manual design of behavioral rules has been the most common approach to control modular robots and self-assembling robots. Letting the robots learn behavioral rules will be the subject of future research, as briefly mentioned in the discussion section: "*Understanding which morphology is appropriate to which task and environment is a problem nature solves over millions of years using evolution. To solve the same problem on the fly, we might be able to rely on ever increasing computing power and advances in evolutionary computation techniques*" (lines 165–168).

Moreover, I agree with the statement (Line 80-82) "Writing the control logic for a composite MNS robot is an entirely new challenge. It is impractical for such logic to take into account every possible morphology, i.e., relative placement of sensors, actuators and body parts." However, In Supplementary Fig. 3, we provide a detailed example of how spatial and temporal actuator coordination happen in an MNS robot. (Line 100-101) does not include such an explanation.

- The relation between the high-level control logic and the low-level instructions was indeed not clear from Supplementary Fig. 3 or its caption. We have updated the figure and its caption to be explicit about which high-level commands are translated into low-level instructions by the individual units.